# Application of Protein in Extrusion-Based 3D Food Printing: Current Status and Prospectus

**DOI:** 10.3390/foods11131902

**Published:** 2022-06-27

**Authors:** Ziang Guo, Muhammad Arslan, Zhihua Li, Shaoyi Cen, Jiyong Shi, Xiaowei Huang, Jianbo Xiao, Xiaobo Zou

**Affiliations:** 1Agricultural Product Processing and Storage Lab, School of Food and Biological Engineering, Jiangsu University, Zhenjiang 212013, China; 2112018015@stmail.ujs.edu.cn (Z.G.); arslanujs@gmail.com (M.A.); lizh@ujs.edu.cn (Z.L.); 2222018002@stmail.ujs.edu.cn (S.C.); shi_jiyong@ujs.edu.cn (J.S.); huangxiaowei@ujs.edu.cn (X.H.); 2Department of Analytical Chemistry and Food Science, Faculty of Food Science and Technology, University of Vigo-Ourense Campus, E-32004 Ourense, Spain; jianboxiao@uvigo.es; 3International Research Center for Food Nutrition and Safety, Jiangsu University, Zhenjiang 212013, China

**Keywords:** animal protein inks, extrusion 3D printing, plant protein inks, personalized food, 3D food

## Abstract

Extrusion-based 3D food printing is one of the most common ways to manufacture complex shapes and personalized food. A wide variety of food raw materials have been documented in the last two decades for the fabrication of personalized food for various groups of people. This review aims to highlight the most relevant and current information on the use of protein raw materials as functional 3D food printing ink. The functional properties of protein raw materials, influencing factors, and application of different types of protein in 3D food printing were also discussed. This article also clarified that the effective and reasonable utilization of protein is a vital part of the future 3D food printing ink development process. The challenges of achieving comprehensive nutrition and customization, enhancing printing precision and accuracy, and paying attention to product appearance, texture, and shelf life remain significant.

## 1. Introduction

Extrusion-based 3D food printing has become one of the most popular molding methods in the food sector in recent years since Cornell University researchers firstly introduced this technology into the food sector [1]. Extrusion-based 3D food printing, also known as fused deposition modeling, was originally used for printing filamentous plastics that can be melted by heating [2]. The 3D food printer continuously extrudes the molten material or paste-like material from the moving nozzle and adheres it to the previously printed raw material layer, and finally completes the printing layer by layer under the guidance of the model data [3]. The technical principle of extrusion printing in the form of a pneumatic, piston drive, or screw drive system is simple, easy to operate, and delivers a low-cost product [4]. The biggest challenges in extrusion-based 3D food printing include printing inks’ properties, printing parameters, and post-processing [1]. The influence of the composition and properties of raw material which include thermodynamic properties, rheological properties, and curing cross-linking mechanism of printing inks on product quality has always been a research hotspot for researchers [5]. The aforementioned properties are particularly important in the extrusion process and self-supporting states. The viscosity (*η*) of the printing inks should be low enough to ensure that the material can be smoothly extruded from the smaller nozzles, and high enough to ensure that the printing inks have sufficient mechanical strength to maintain the stability of the printing products [6]. Previous reports claimed that the various food materials such as polysaccharides [7], proteins [8], and fats [9] met the properties described above.

Protein is a polymer compound formed by combining one or more polypeptide chains consisting of amino acids in a specific way [10]. Eukaryotes possess approximately 20 standard amino acids, forming biodiversity based on protein materials [11]. Protein-based 3D printing inks offer a wide range of applications owing to the superiority of protein materials [12]. The application of protein-based 3D printing inks includes food production [13], drug design [14], biological applications [15], and smart packaging [16], among others. Food researchers focused on the sources of new raw materials, nutritional qualities, post-processing, texture, and taste of the final product as 3D food printing technology progressed. Furthermore, the edible protein modified the texture and taste of the product owing to its various sources and internal component structures, and created beneficial support or nutritional benefits of the printed model.

Protein inks have a variety of effects on printing results in 3D food printing. Printability, printing parameters, rheological characteristics, equipment, and printing materials have been the primary study topics in 3D food printing protein research in core journals from Web of Science over the last 5 years, as shown in Figure 1. The main research topics were focused on 3D printing equipment, printing materials, and printing parameters. In addition, the rheological properties of inks have been emphasized in that period of time, and the use of protein as a stabilizer for high internal phase emulsions for 3D printing inks has been a research hotspot for nearly a year. This is a huge step forward in the individuation and functionalization of 3D printed food owing to its advantages of embedding and controlled release [17].

Protein is mainly classified as plant protein and animal protein according to the different sources. Protein played a special role in 3D printed food by providing support, nutrition, and printability. Physical and chemical qualities, as well as extrinsic elements such as additives, post-processing, and printing conditions, have a significant impact on proteins. To the best of our knowledge, systematic reviews on this topic have not been published. Therefore, the aim of this review article was to collect and analyze information regarding the function and influencing factors of protein in extrusion-based 3D food printing. Moreover, the application of proteins in 3D printing inks and critical views on the future of 3D food printing were discussed.

## 2. The Function of Protein in 3D Food Printing

Protein is an important raw component for printing ink and plays an essential role in extrusion-based food 3D printing. It is necessary to understand the relationship between the characteristics of the material and the end-product to construct a more complete and acceptable 3D printing product. A schematic diagram of major analytical steps considered for 3D printed protein-based products is presented in Figure 2.

### 2.1. Printable Function

Printing inks were applied in extrusion-based 3D food printing through blending and depositing self-supporting layers of printing inks. The rheological properties of the printing ink were a factor that cannot be ignored [18]. In extrusion-based 3D food printing, the printing inks must be readily squeezed out via the nozzle and have adequate mechanical strength to maintain the construction [19]. The addition of protein helped inks with strong shear thinning behavior and a rapid shear recovery property. Shear-induced elongation or stretching of protein molecules improves sliding between molecules and further reduces molecular entanglements, lowering *η* and frequently resulting in shear-thinning behavior [20].

The extracted gelatin from the structure of collagen fibers was a common protein 3D printing ink thanks to its excellent texture and gel properties [21]. Chen and coworker reported that the gelatin was added to soy protein isolate (SPI) to extrude out and support the deposited layers [22]. Another study reported that the mixtures of SPI and gelatin could print stable and precise geometries. The acquired results revealed that when gluten protein was compatible with water, *η* of the gluten protein in the dough was increased. The coagulation function in the post-processing process was increased, making it an ideal 3D printing ink [21]. The increase of *η* improved the uniformity of 3D printed products and their subsequent molding. The non-Newtonian shear thinning behavior of surimi gel was demonstrated by the reduction in *η* with an increasing shear rate. The storage modulus (G’), loss modulus (G”), and loss tangent (tan δ) represent the viscoelastic behavior of surimi materials, and larger G’ values more greatly indicate the printed models’ stability. For all surimi samples, the G’ was always higher than the G”, with tan δ less than 1 [23]. When tan is less than 1, the material takes on the characteristics of a flexible gel with a gel-like microstructure. The addition of κ-carrageenan increases values of G’ and G”, which was conducive to the maintenance of the print product shape. However, the synergistic relationship between many exogenous substances and surimi protein is unclear and needed to be further clarified. It is also worth looking into using beef or poultry for 3D printing to find the best recipe and save money on raw materials. The mechanism by which the mixture of whey protein isolate and maltitol improves the stability of the formula is described as follows [24]. The higher the protein content, the stronger the bonds between proteins and the stronger the network formed by the migration of water to the solid protein particles. The observation of the microstructure could also have confirmed the presence of a continuous and uniform structure, which also explained the reason why the high protein content would increase the yield stress, *η*, and consistency index of the printing matrix.

High internal phase emulsions as 3D printing inks offer a new research direction in the field of food science and engineering. High internal phase emulsions stabilized using proteins displayed self-supporting stability offered by the highly flocculated emulsion droplets [25]. The stabilizer between the two phases played a critical role in the overall stability of the printing inks. This stability is mainly due to the formation of networks of protein particles in the continuous phase. The stabilization mechanism depends on appropriate force of non-adsorbed protein particles in the continuous phase to form the three-dimensional viscoelastic particle network and prevent the oil droplets from moving [26]. Therefore, it had a significant impact on the stability and shape accuracy of 3D printed products. Previous studies have also reported the use of rice protein [27], pea protein [28], gelatin [25], and cod protein [29] in this application. In conclusion, the printability of proteins should be the basis of 3D food printing research in the future.

### 2.2. Amino Acid Supplements Function

Protein is a potential printing ink for producing 3D printed food that meets the specific needs of consumers such as the elderly, pregnant women, children, the sick, and athletes in the field of food processing [30]. Protein is a rich source of essential and non-essential amino acids and offers a variety of health benefits for human health as printing inks.

Meat is an important source of protein in the human diet and contains various essential amino acids. According to a previous study, animal proteins have a good influence on skeletal muscle mass due to their favorable effect on body protein synthesis [31]. In another work, pork modified with edible gel was printed to provide energy for dysphagia patients [32]. The surimi protein contains an abundant amino acid with shorter myofibrillar proteins that are easier to digest in the body as compared to meat. The study investigated the effects of ultrasound-assisted and water immersion thawing on the protein structure of silver carp surimi [33]. The acquired results suggested that thawing surimi at a higher ultrasonic frequency (80 kHz and 100 kHz) reduced the damage to the secondary and tertiary structure of myofibrillar proteins. The hydrolysis capacity of a protein is determined by its structure, which also influences the digestibility of that protein [34]. Plant protein can be a meat protein alternative when processing costs and amino acid types are considered. The two most prevalent sources of plant protein are soy and pea proteins [35]. Another study reported the use of SPI as a nutrient in food printing along with mixtures of pumpkin and beetroot [36]. The pea protein concentrate or SPI was used in place of part of the porcine plasma protein to keep an overall biopolymer content within the dough [37]. In comparison to animal protein, the results showed that lower protein digestibility corrected the amino acid score (PDCAAS) values. PDCAAS readings for pea range from 0.33 to 0.75, and for soya from 0.54 to 1.00, depending on preparation technique [31]. A combination of different proteins should also be employed for 3D food printing to achieve a nutritionally balanced amino acid profile. This is an important step forward in the direction of personalized products and customized nutrition requirements. In addition, single-cell protein is also an important nutritional component that cannot be ignored in food 3D printing. The single-cell protein is a rich source of various essential amino acids such as lysine and methionine which are not present in sufficient proportions in most animal and plant sources [38]. A study explored the potential of food 3D printing to incorporate microalgae (*Chlorella vulgaris* and *Arthrospira platensis*) in cereal snacks [39]. The results of the study suggested that the composition and structure of proteins from different sources are diverse.

## 3. Influencing Factors of Protein Function in 3D Printing

The function of protein in 3D printing is influenced by physical and chemical properties of the material (Figure 3). These factors have a significant effect on promoting the printing ink extrusion from the nozzle, deposition molding, and the stability of 3D printing products. Experimental and optimization conditions involving influencing factors are listed in Table 1 and Table 2.

### 3.1. pH of the Printing Ink

The protein contained various negatively and positively charged amino acid functional groups. The charge of the protein-polymer was largely dependent upon the isoelectric point (pI) of the protein and the pH of the solution [40,41]. The adjustment in the pH of the protein solution could change its structure, which might further alter the printability of the printing ink. The study reported the use of protein-stabilized high internal phase emulsions for food 3D printing. The pH of the continuous phase (cod protein solution) was far from the pI and falls in the alkaline range (pH = 10), creating a stable printing system [29]. This could be due to a higher absolute value of protein potential (negative value), higher electrostatic repulsion of protein molecules, a lower hydrophobic force of surface groups, and changes in protein structure, all of which contribute to the formation of a stable three-position network structure under alkaline conditions [42]. This applies to rice protein as well [27]. In addition, the protein molecule aggregated at the pI, and the *η* value is the smallest in the presence of both charges. Aggregation degrees of proteins can affect liquid-based printing processes’ gelation-forming mechanisms. Similar charges that repel each other extend the distance between molecules and increase the *η* value of printing inks. For instance, pH is an important factor in the solution-gel conversion process of dairy products. A study reported that the milk gels of pH 4.8 and 10% protein were the optimal condition for 3D printing. The lowest *η* of protein was observed between pH 5.2 and 5.4, which was close to pI. The gelation process can also reduce the temperature dependence under these conditions [43].

### 3.2. Functional Carbohydrates

Food proteins and functional carbohydrates (carrageenan, xanthan gum, pectin, and dietary fiber) were mixed and deposited in the 3D food printing process to improve extrusion and molding conditions. This could be due to the addition of polysaccharides to a printing ink system which causes the state of bound water to change to free water, resulting in the interaction of molecular attraction forces and repulsion forces, which changes the molecular structure of the polysaccharide or protein and, as a result, affects the rheological properties of the printed material [44]. Another study revealed that the whey protein damaged the gel structure of Konjac gum and diminished the hydrophilic ability. This could be due to the whey protein network hindering free water migration, or the increased number of whey protein molecules impeding free water migration [45]. Phuhongsung and coworkers confirmed that high concentrations (3%, *w*/*v*) of κ-carrageenan improved the printability in the SPI gel network owing to the aggregation of double helices inducing the formation of an appropriate network structure [46]. Another study was designed to explore multi-scale ink produced by the separation of gellan gum and whey protein isolates through the separation phase [47]. The results of the study reported that the increase of the gellan gum amount improved the shear-thinning behavior and shear recovery ability of the ink. This phenomenon was attributed to the *η* value of gellan gum.

### 3.3. Enzymes

The addition of enzymes to protein printing inks significantly affects the self-internal structure. Transglutaminase (TGase) is a widely utilized enzyme that causes protein gels to form. It created the protein gel by inducing an acyl-transfer reaction between the γ-carboxamide group of the remaining glutamine and the -amino group of lysine in other proteins, resulting in intermolecular or intramolecular cross-linking (ε-(γ-glutamyl)-lysine cross-linking) [48]. Dong and coworker revealed that the *η* of surimi containing 0.2% and 0.3% (*w*/*w*) TGase facilitated the extrusion of surimi from the nozzle. The acquired results revealed that the hardness, cohesion, and robustness of the printed gels increased as the TGase level rose from 0 to 0.4 percent. In addition, it was also observed that the gel structure changed from irregular to tight uniform network [49]. Moreover, the storage stability and printing performance of TGase cross-linking gelatin to stabilize the high internal phase emulsions were outstanding. Therefore, TGase cross-linking duration is a key element determining the qualities of the end-product [25].

### 3.4. Heating Treatment

The rheological characteristics of protein inks are heavily influenced by temperature. The protein denaturation caused by the higher temperature increases the amount of exposed hydrophobic groups that promote hydrophobic-type interactions as well as the number of reactive sites for covalent bonding [50]. A study found that the ink system’s (SPI-strawberry:2–1) printing accuracy and self-supporting performance were considerably enhanced by low power (70 W) microwave treatment [51]. Another study also reported the heating power level between 50 and 80 W. Temperature management of protein solution-phase transitions may be used to meet 3D printing requirements [46]. In another study, temperatures of the sol-gel transition of skim milk retentates at various protein (8–12%) and pH (4.8–5.2) levels were determined [43]. The extra improvement in gel firmness might be due to the simultaneous heat treatment of the acid gels. Moreover, temperature-induced phase separation and compacting were recognized as a competitive process that resulted in firm but coarse gel networks [52]. The curing process is an important step for meat product models. Surimi is one of the most fascinating situations owing to the substrate’s complexity. Surimi gelled during heat processing and could not be printed owing to a rise in cathepsin activity [53]. However, surimi material exhibits outstanding shear thinning behavior under appropriate microwave intensity and in the presence of TG. Surimi thinning was observed when microwave power was less than 60 W/g. After printing at 40 W/g and 50 W/g, solid gels with greater form integrity and large protein aggregates emerged. The author also reported that hydrogen bonds and ε-(-Glu)-Lys are the major factors in maintaining molding quality, and microwaves activate TGase to encourage the process of self-gelation [54]. As a result, temperature is an excellent approach to transition from a liquid to a solid state, as well as to increase the molding quality of printing ink during printing progress.

## 4. The Application of Protein in 3D Food Printing

Dietary protein is mainly divided into animal protein and plant protein [55]. Accurate and sufficient protein intake is of great importance for people with special needs for protein. Therefore, the customized food with 3D printing technology highlights the value and significance of its existence.

### 4.1. Animal Protein

#### 4.1.1. Whey Protein

Whey protein is a kind of protein derived from milk that has high nutritional value. Whey protein aids digestion and absorption in the intestine and contains a number of active substances [56,57]. Whey protein is one of the high-quality protein supplements for humans. It is frequently utilized in food production and processing owing to its nutritional value and functional property. Whey protein is a globular protein with a distinct secondary and tertiary structure. Thiol-disulfide exchange reactions occurred as a result of heating at 70 °C. As a result, both the whey and casein protein underwent denaturation or aggregation reactions [58], and formed a 3D network structure [59]. Du and his colleagues designed a study to use a sort of hybrid gel of Konjac and whey protein as a 3D printing ink [45]. The results of the study revealed that the rheological properties (G’ and *η*, etc.) and texture (hardness, springiness, etc.) of Konjac gel were improved owing to whey protein supplementation. This was because the mixture created a novel gel structure with higher density. The whey protein at a concentration of 20% produced an optimal gel effect. Daffner and coworkers investigated the effect of formulation parameters (casein content, whey protein content, and pre-processing (pH-temperature)) of casein whey protein suspensions [52]. They found that changes in the properties of the casein caused the transition temperature sol-gel by changing the pH of the thermal processing process, which in turn increased the aggregation rate. The aggregation rate was closely related to the adaptability of 3D printing. Additionally, it also brought up new possibilities for researching more complicated printing ink systems and formulating personalized and nutritious food. The animal-protein-based printing inks have been summarized in Table 1.

#### 4.1.2. Egg Albumin

The most abundant protein component in eggs is egg albumin, which contains 54% of egg white protein (EWP) [56,60]. EWP is a food additive that performs a variety of functions, including foaming, emulsification, heat-set gelation, and binding adhesion [61]. EWP has become a potential 3D printing ink for the development of new types of food owing to its heat-induced edible gel characteristics [56]. Liu and coworkers studied egg albumen protein powder with 80% EWP content as the raw material and explored the influence of the addition of EWP on the rheology, lubrication, texture, and microstructure of the mi attributes mixture system [62]. Results of the study reported that printing ink containing 5% EWP was most suitable for 3D printing. They also revealed that adding a certain concentration of EWP can improve the hardness and elasticity of the gel sample, and serve as a good auxiliary material for 3D printing inks. Likewise, another study proposed an optimization plan for a new 3D printed food formula containing an EWP system [63]. In 250 mL of total deionized water, the best solution ratio was: EWP (12.98 g), sucrose (8.02 g), cornstarch (19.72 g), and gelatin (14.27 g). EWP can be widely used in food 3D printing due to its excellent gel properties. However, the potential for this protein to produce allergic reactions must be considered. Therefore, it should be employed with reasonable matching to achieve a protein balance [56,64].

#### 4.1.3. Gelatin

Gelatin is produced by hydrolyzing collagen, and has swelling and gelling properties. Flexible single random coils were initially generated after the dry gelatin was fused into heated water at 40 °C. The polypeptide chain in the connecting region was restored to the triple-helix-like structure of collagen after cooling, resulting in final gelation [4,65]. Chen et al. explored the application of a mixture of gelatin and SPI as 3D printing ink in food [22]. The findings of the study revealed that SPI extruded and supported the deposited layer with the help of gelatin. The mixture of SPI and gelatin can print stable and precise geometries. Strother et al. investigated the effect of gelatin on the sensory and textural properties of carrot puree. The acquired results revealed that adding gelatin to the printing ink would improve the hardness of the ink. The author suggested that the filling percentage, nozzle diameter, flow rate, and nozzle height should be considered for the better texture of the final product [66]. Another study reported the application of thermally stable emulsions in 3D printing [25]. The stability of the emulsion was largely improved by TGase cross-linked gelatin. The finally obtained high internal phase emulsions have excellent storage stability and printability.

#### 4.1.4. Surimi

Surimi is prepared from cleaned fish meat that is free from impurities such as sarcoplasmic protein, lipids, blood, enzymes, and other contaminants [67,68]. The unique ability of surimi to produce gels stems from the presence of a salt-soluble myofibrillar protein [69]. Previously, various studies have reported surimi to be an appropriate material for extrusion-based 3D food printing [49,70,71]. However, sodium chloride is one of the important factors influencing the rheological properties of surimi gel. It is frequently used to dissolve fibril in protein to induce folding. The rheological properties of the gel fluctuate in response to changes in sodium chloride concentration [72]. A previous study verified that the surimi printing ink containing 1.5% sodium chloride can produce mechanical properties suitable for 3D printing [71]. Another study reported that addition of microbial TGase as an additive significantly improved the extrudability of surimi [49]. It can also accelerate the formation of gel under the action of microwave heat induction [54]. Surimi could be a vital printing ink for future 3D printing as a protein-rich raw material.

#### 4.1.5. Insect Protein

Nutritionists are increasingly paying attention to insects as significant carriers of nutrients such as protein and minerals. The protein level (50–85%) and digestibility (75–98%) of insects may be superior to those of plant and animal protein [73]. However, most humans are reluctant to eat insects [74]. Insect protein as a component of 3D printing ink is one of the sustainable ways to promote their consumption. The study designed a cylindrical snack made of dough containing yellow mealworm (*Tenebrio molitor*) protein [74]. The study’s findings found that the addition of different levels of *Tenebrio molitor* significantly modified the dough’s printability and morphology and microstructure characteristics of raw snacks. In another study, yellow mealworm larvae were blended with soft candies to make icing for the top cake decoration [75]. Currently, there is very little study in this field. Therefore, researchers should explore the sensory properties of printing inks of different concentrations of protein in the 3D printing process while exploring new insect proteins.

#### 4.1.6. Other Protein

Animal protein is the most prevalent source of protein for humans, which comes in a variety of forms. The study was designed to investigate the printability of dough with different pig plasma protein content [37]. The findings of the study revealed that only dough with pig plasma protein content between 42.5% and 47.5% weight could be printed successfully. Likewise, in another study, characteristics of white shrimp (*Litopenaeus vannamei*) were improved by adding 6% cross-linked starch, making it more appropriate for 3D printing [76,77].

### 4.2. Plant Protein

#### 4.2.1. Soy Protein

SPI is a rich source of protein in the human diet, containing both essential and non-essential amino acids [78]. The SPI gel has superior *η* values and rheological properties, making it unsuitable for single-material printing [79]. The study reported that the mixture of SPI and hydrogel (gelatin) could be a promising 3D printing ink, producing stable and accurate geometric shapes while maintaining nutrition [22]. The addition of salt (KCl, CaCl_2_, NaCl, CaSO_4_, MgCl_2_, and NaC_6_H_7_O_6_) was a simple way to change the properties of SPI gels, causing rapid protein aggregation and gelation [79]. Likewise, a study was designed to optimize the formulation of xanthan gum and NaCl concentration for SPI printing ability. The acquired results revealed that the solution containing xanthan gum and NaCl at a concentration of 0.5 g/30 g and 1 g/100 mL print perfect shapes, respectively. The results also reported that exogenous stimulation has a variety of effects on printing SPI-containing inks [79]. Fan et al. studied a new microwave (low power)-salt co-pretreatment on the SPI-strawberry 3D printing system [51]. Microwaves achieved ink printability at low power and improved product acceptability. The addition of salt promotes the formability and stability of 3D printed products. Another study also reported the mixture of SPI, κ-carrageenan, and spices to be used as a printing ink [46]. Moreover, stimulation at different pH values can change the color, texture, and flavor of SPI as a 3D-printed product [36]. The plant-protein-based printing inks have been summarized in Table 2.

#### 4.2.2. Pea Protein

Pea protein includes all amino acids except methionine, and its anti-nutritional factors are substantially lower than those found in other plant proteins, making it easier to absorb by the body [56]. The appropriate increase of pea protein in the diet can help optimize food nutrition, improve texture, and increase product stability [80]. The pea protein delivers superior raw materials for the elderly, children, athletes, or patients to provide high-quality 3D printed products. The study investigated the effect of pea protein on the printing properties of potato starch 3D printing ink [81]. The findings of the study revealed that the smoothness, thermal properties, and structural properties of the ink were greatly improved with the addition of pea protein. Another study delivers the best printing conditions by analyzing the rheological, thermodynamic, and texture properties of different alginate and pea protein ratios [82]. Pea protein mimics the feel of fat, which is a major step forward in replacing fat with low-calorie ingredients. The intake of high-quality protein greatly lowers the incidence of obesity, atherosclerosis, and other diseases [83]. The addition of a proper amount of pea protein into the printing ink can improve the stability of the structure and balance the nutritional value of the printed food.

#### 4.2.3. Gluten Protein

Gluten is an important component of dough and a crucial factor to consider while 3D printing dough. The differences in dough properties, such as moisture, protein, or starch content are reflected during the printing and post-processing stages. The hydration in the dough system will promote the formation of a continuous network of gluten macromolecules [84]. However, addition of higher amounts of water resulted in gluten chain fracture and formed a discontinuous network. The fracture was mainly due to the breaking of disulfide bonds that connected polypeptide subunits and subsequent depolymerization of gluten [85,86]. The external stimulation (microwave, baking, etc.) can reduce the moisture content of the dough, make the dough gelatinize, strengthen the gluten network structure, and thus increase the viscoelasticity of the dough [87,88].

#### 4.2.4. Other Protein

The various other plant proteins which have good printability and nutritional value have also been applied in 3D printing. The addition of oat protein and fava bean protein improved the printability of printing materials and prevented syringe clogging [5]. The study reported that the average diameter range of peanut protein isolate printing ink fluctuated little and the average diameter was low, indicating that the printing accuracy of peanut protein isolate was the best [89]. Likewise, other studies realize the control of the three-position structure by adjusting the printable ingredients (20% glycerol) based on zein, and opening up the field of processing absorbable printing parts [90,91].

## 5. Future Outlook of Protein Inks in 3D Food Printing

The fundamental task of 3D-printed food is to realize the comprehensiveness or customization of nutrition. These tasks enable a guarantee of rigorous product quality and precise control of nutrition. People who require a high-quality and easily digestible protein supplement (children, athletes, elderly, sick, and pregnant women) prefer high-quality and easily digestible protein as the composition of printer inks. Excessive or insufficient nutrition for the particular population can be reduced or eliminated using 3D printing technology. The long-term sustainability of animal and plant protein consumption is a continuing concern as their intake directly or indirectly determines the health of humans. The researchers should accurately control and pay attention to the purity of printing inks. It is worth mentioning that some insect protein can also be utilized as an alternative ingredient for consumption in response to shortages. This protein replacement strategy not only made utilization of high-absorption insect protein possible but also reduced residues and achieved sustainable development. Plant protein has sparked a lot of interest among nutritionists as a possible animal protein alternative. Plant protein is a good source of dietary protein and has a longer shelf life and better health advantages than animal protein. Furthermore, plant protein satisfies the nutritional needs of vegetarians and has the potential to become a component of the rapidly expanding meat alternative market. However, the effective and reasonable utilization of protein is a vital part of the future 3D food printing ink development process.

Printing quality and accuracy, as well as essential ingredients to boost customer interest and desire, are critical to the use of 3D food printing based on balanced food nutrition. The protein is rarely printed as a single material. The protein printing inks should be explored in terms of formula ingredients to improve printing accuracy. The material properties and printing controlling parameters must be carefully monitored for a better final product. The self-support and stackability are controlled by protein inks combined with additional additives such as hydrogels, enzymes, starches, glycerin, and plasticizer in the protein matrix. The printing control parameters and post-processing can be adjusted according to the role of the raw protein to realize the potential for precision and customization in the structure of the printed product.

In addition, maintaining the appearance and texture and extending the shelf life of printed products were the key issues. The popularity of 4D printing has prompted a more in-depth examination of product appearance factors such as color and shape. The introduction of raw protein materials may lead to changes in the density and other internal structures of the printed products, thus giving rise to 3D printed food of new texture. These challenges and opportunities for protein-based printing inks need to be explored in future studies.

## 6. Conclusions

Extrusion-based 3D food printing is one of the most common ways to manufacture complex shapes and personalized food. Protein as a functional 3D food printing ink has always had an amino acid supplements function and printable function in 3D printing products in terms of personalized ingredients and food design. The printing accuracy and product appearance of this type of printing ink were easily affected by various factors such as pH, hydrogel, enzymes, and heating-processing. Despite advances in 3D food printing technology, challenges such as achieving comprehensive nutrition and customization, rational protein resource utilization, enhancing printing precision and accuracy, and paying attention to product appearance, texture, and shelf life persist. These obstacles based on protein raw materials as a functional 3D food printing ink should form the basis of future research exploration.

Printable and nutritional function of protein are important in 3D food printing;Physical and chemical properties of protein have considerable impact on the final outcome;Animal and plant source protein as a function of 3D food printing ink is discussed;Robust extrusion-based 3D printers are needed to produce personalized foods

## Figures and Tables

**Figure 1 foods-11-01902-f001:**
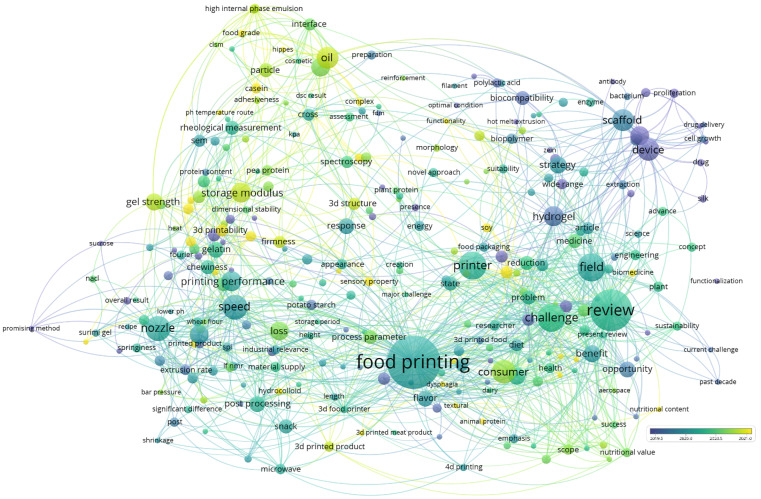
Analysis results of “food 3D printing protein” hotspots in recent years.

**Figure 2 foods-11-01902-f002:**
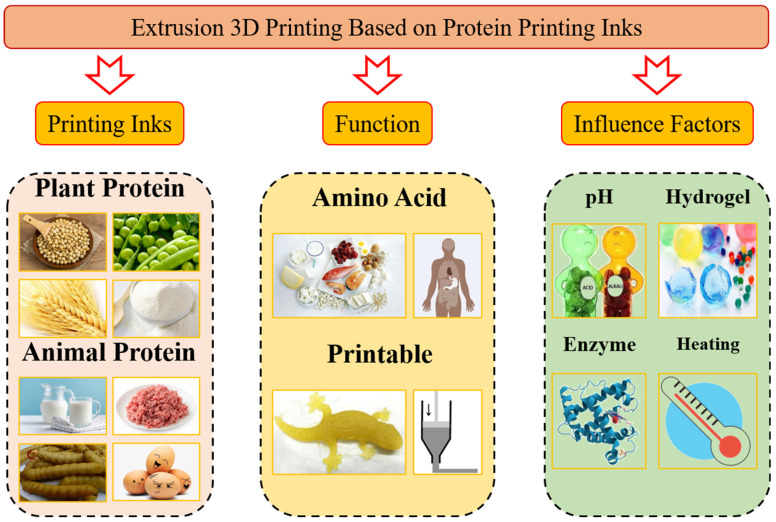
Schematic diagram of major analytical steps considered for 3D printed protein-based products: printing inks, function, influencing factors, and applications.

**Figure 3 foods-11-01902-f003:**
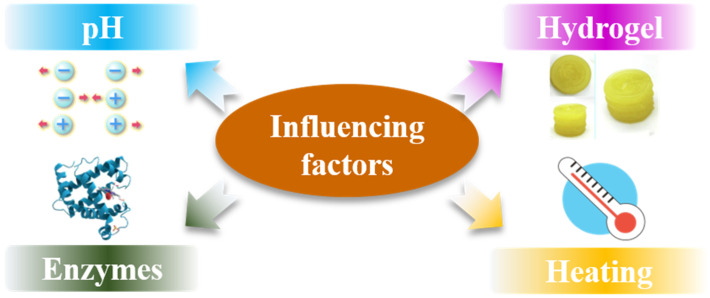
Overview of the influencing factors for protein function in 3D food printing.

**Table 1 foods-11-01902-t001:** Animal-protein-based printing inks.

Category	Other Materials	Experimental Conditions	Optimal Conditions	Actual Application Results	Reference
Whey protein	Casein	Acid environment, heating treatment	pH: 4.8–5.0; T: 80 °C	Mixture suspension is printable and forms a stable gel	[1]
	Konjac	Continuous mixing at heating treatment	T: 85 °C	Gelling property performed best as an additive (20%)	[2]
	Canola oil, corn starch	Continuous mixing at room temperature	-	Excessive whey protein impaired the printing inks’ printability	[3]
	NaCl, fat, protein	Severe mixing at neutral condition	pH: 7	Effect of whey protein as printing inks and additives on printing results was explored	[4]
	Starch, beeswax	Continuous mixing at heating treatment	T: 90 °C	As a printing ink component to meet the nutritional function of the product	[5]
Egg albumin	Rice flour	Severe mixing, heating conditions	T: 70 °C	Egg yolk has a higher binding capacity to starch compared to egg albumin, and the effect of 3D printing was good	[6]
	Sesbania gum	Continuous mixing at heating condition	T: 75 °C	The 0.1% SG protein gel mixture structure was stable, there was no covalent bond between them	[7]
	Gelatin, sucrose cornstarch	Severe mixing, heating conditions	pH: 6.5, T: 55 °C	EWP can improve the hardness and elasticity of the gel sample	[8]
	Gelatin, sucrose cornstarch	Severe mixing, heating conditions	pH: 6.5, T: 55 °C	An optimization plan for a new 3D printed formula containing EWP system	[9]
Gelatin	TGase, soy oil	Mixing at heating conditions	pH = 5.0, T: 60 °C, TGase: 2 mg/mL	TGase cross-linking of gelatin effectively improved the thermal stability of HIPEs.	[10]
	Zinc oxide, clove essential oil	Extrusion at room temperature		Gelatin/zinc oxide/clove oil nano-packaging was developed for food preservation.	[11]
	Pureed carrots	Mixing at heating conditions	T: 45 °C	Samples made with gelatin were the hardest (texture profile analysis) product	[12]
	TGase	Mixing at heating conditions	T: 40 °C TGase: 5%	Preheating of gelatin improves its printability with TGase	[13]
Surimi	NaCl	Mixing at room temperature	NaCl: 1 g/100 g	Better prediction obtained for multiple rheological parameters by LF-NMR.	[14]
	NaCl	Mixing at room temperature	NaCl: 3%	SEM showed that 3% salt was suitable for 3D printing ink using fish surimi.	[15]
	MTGase	Mixing at room temperature	MTGase: 3%	Microbial TGase impacts 3D printability and extrudability of surimi.	[16]
	TGase	Mixing, microwave printing	TGase: 5 U/g, power: 50 w/g	MW3DP and TGase can be used for 3D printing of heat-induced gel food.	[17]
Yellow mealworm	Dough	Mixing at heating treatment	-	Changed the printability of the dough, improving the texture, digestibility, and microstructure of snacks	[18]
Pig plasma protein	Glycerin, dough	Mixing at room temperature	-	Dough with pig plasma protein content between 42.5 and 47.5% weight could be printed successfully.	[19]

Note: “T” stands for temperature. The experimental conditions and optimal conditions are selected from the influencing factors involving proteins. The values of the optimization conditions are the parameters of printing ink formation.

**Table 2 foods-11-01902-t002:** Plant-protein-based printing inks.

Category	Other Materials	Experimental Conditions	Optimal Conditions	Actual Application Results	Reference
Soy protein	Xanthan gum, NaCl	Severe mixing and microwave	pH:7; mixing: 6400 rpm, 5 min; microwave:100 W, 5 min	SPI gel with xanthan gum and NaCl solution at 1 g/100 mL could be successfully printed	[20]
	κ-carrageenan, vanilla flavor	Severe mixing and heating treatment	Mixing: 6400 rpm, 5 min; T:70 °C	SPI gel made with 3% (*w*/*v*) carrageenan was the most suitable for 3D printing	[21]
	Pumpkin, beetroot	Severe mixing and microwave	Mixing: 6400 rpm, 5 min; microwave: 100 W, 5 min	The best printing results were obtained when stimulated with pH = 6	[22]
	Strawberry	Mixing and microwave	microwave: 70 W	Salt and microwave treatment improved the printing accuracy and self-supporting performance of the ink system	[23]
Pea Protein	Alginate	Mixing and heating treatment	T: 43 °C	Alginate solution 80% and pea protein solution 20% were found to be the most suitable for 3D printing	[24]
	Potato starch	Mixing	-	Pea protein improved the texture, thermal property, and structural properties of the ink	[25]
Gluten protein	Milk, fat	Mixing	-	Cookie dough formulations with reduced sugar content were more printable	[26]
	….	Mixing and heating treatment	T: 55 °C	The heating condition affected the protein structure and improved the printing effect of printing ink	[27]
Peanut protein	Hawthorn powder	Mixing and storage	Storage temperature and time: 4 °C, 12 h	The mixture has good 3D printing properties and applies to other fruit and vegetable inks	[28]
Oat protein & fava bean protein	-	Severe mixing	-	Mixture improved the printability of printing materials, but the forming effect was defective	[29]
Zein	-	Mixing	-	Realize the control of the three-position structure by adjusting the printable ingredients	[30]

Note: “T” stands for temperature. The experimental conditions and optimal conditions are selected from the influencing factors involving proteins. The values of the optimization conditions are the parameters of printing ink formation.

## Data Availability

The data are available from the corresponding author.

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
