# Peer review of "Application of Protein in Extrusion-Based 3D Food Printing: Current Status and Prospectus"

_foods, 2022, doi:10.3390/foods11131902_

Round 1

Reviewer 1 Report

I reviewed the manuscript entitled, Application of protein in extrusion-based 3D food printing: Current status and prospectus. This review covered the role of proteins in 3D food printing with focused discussion. Moreover, this topic filled an identified research gap addressing the protein in 3D printing. Overall, the review is interesting, comprehensive and with relevant references, and established the need for future studies on 3 D food printing using proteins. 

Abstract: Authors should also conclude the review findings and future recommendations.

Figure 1 quality is extremely bad. Cannot even read to max zoom of 500 %. Author should produce high quality figure.

3. Influencing factors of protein function in 3D printing:  table containing information on influencing parameters, such as how much protein charge is good for food printing, carbohydrates example or role; enzyme; temperature range etc should be included.

References are not according to journal format.

Author Response

Manuscript ID: foods-1769054

Application of protein in extrusion-based 3D food printing: Current status and prospectus

We are grateful to the editors and anonymous reviewers for their valuable comments and suggestions, which helped us to improve the quality of the Review article. We have revised the reviewer’s comments carefully. A revised version of our manuscript (marked with red color) has been uploaded in the online system.

Reviewer 1

I reviewed the manuscript entitled, Application of protein in extrusion-based 3D food printing: Current status and prospectus. This review covered the role of proteins in 3D food printing with focused discussion. Moreover, this topic filled an identified research gap addressing the protein in 3D printing. Overall, the review is interesting, comprehensive and with relevant references, and established the need for future studies on 3D food printing using proteins.

Response: Dear Reviewer, Thank you for a greater insight into the review article. We have addressed the comments/suggestions provided and revised the article accordingly. The point wise reply of each question/query is provided below.

Question 1: Abstract: Authors should also conclude the review findings and future recommendations.

Response: Dear reviewer, thank you for this good advice. Based on your suggestion and original content, abstract conclude the review findings and future recommendations.

Question 2: Figure 1 quality is extremely bad. Cannot even read to max zoom of 500 %. Author should produce high quality figure.

Response: We would like to state in the kind honour of the respected reviewer that Figure 1 has been replaced with high quality figure (Please check with Figure 1).

Question 3: 3. Influencing factors of protein function in 3D printing: table containing information on influencing parameters, such as how much protein charge is good for food printing, carbohydrates example or role; enzyme; temperature range etc. should be included.

Response: Dear reviewer, thank for your kind suggestion. According to your opinion, we have added relevant parameters (experimental conditions and optimal conditions) related to influencing factors into the tables. (Please check with Table 1 and Table 2).

Question 4: References are not according to journal format.

Response: Dear reviewer, the reference section of the manuscript has been checked and revised according to the journal format. Thank you for helping us to improve the quality of our review article.

Reviewer 2 Report

Comments on the content in the introduction - I perfectly understand the importance of color in making decisions (purchase, eating) - I perfectly understand the importance of acceptable and approved dyes and their influence on the properties of masses sent for 3D extrusion, but I think that too much space has been devoted to the topic in this chapter dyes. Of course, I do not take into account the importance of this topic, but maybe the authors should consider separating a subchapter on this topic.

in my opinion chapter 2.2. Nutritional function - not fully discussing the nutritional value of the proteins used - focuses on the diversity of the protein raw material and its properties. There is no word about changes taking place in polypeptide chains, changes in the availability of proteins. It suggests either to complete the content or to correct the chapter

3.1. Protein Charge - in my opinion, the next chapter is not fully titled - the title indicates the types of proteins, their origin, and the content concerns rather interactions and changes depending on physicochemical factors - temperature, ph

3.4. temperature - giving such a title indicates some border values of changes that occur in various proteins, which is quite general. The chapter is too generally written in analytical and temperature processes, there is no concept of "lower", "higher" temperature, if it is higher / lower but than a specific one, e.g. 40 degrees C.

suggests for the parentage 2.2. and consider all 3 to give new titles more relevant

Author Response

Manuscript ID: foods-1769054

Application of protein in extrusion-based 3D food printing: Current status and prospectus

We are grateful to the editors and anonymous reviewers for their valuable comments and suggestions, which helped us to improve the quality of the Review article. We have revised the reviewer’s comments carefully. A revised version of our manuscript (marked with red color) has been uploaded in the online system.

Reviewer

Dear Reviewer, Thank you for a greater insight into the review article. We have addressed the comments/suggestions provided and revised the article accordingly. The point wise reply of each question/query is provided below.

Question 1: Comments on the content in the introduction - I perfectly understand the importance of color in making decisions (purchase, eating) - I perfectly understand the importance of acceptable and approved dyes and their influence on the properties of masses sent for 3D extrusion, but I think that too much space has been devoted to the topic in this chapter dyes. Of course, I do not take into account the importance of this topic, but maybe the authors should consider separating a subchapter on this topic.

Response 1: Dear reviewer, thank you for your kind suggestions. In the introduction, we emphasized the development process of extrusion-based 3D food printing, basic information related to proteins, and the application and development trend of protein printing inks. This fully demonstrates that proteins played a special role in 3D printed food by providing support, nutrition, and printability. Therefore, it was necessary to discuss the application of protein in 3D printing ink, and put forward some critical views on the future of 3D food printing.

Question 2: In my opinion chapter 2.2. Nutritional function - not fully discussing the nutritional value of the proteins used - focuses on the diversity of the protein raw material and its properties. There is no word about changes taking place in polypeptide chains, changes in the availability of proteins. It suggests either to complete the content or to correct the chapter.

Response 2: Dear reviewer, thank you for this good advice. We have re-reviewed section 2.2 and founds the reviewer suggestion valuable. This part mainly explains that protein was used as printing ink to become 3D printed food, which provides amino acids for consumer. Therefore, we have revised the title of this section to "Amino Acid Supplements function". The specific reasons are as follows: Animal protein as a high-quality protein has a good influence on skeletal muscle mass owing to its rich amino acid content. Surimi protein contains an abundant amino acid with shorter myofibrillar protein that are easier to digest in the body. Moreover, considering the comprehensiveness of amino acid types, plant protein and microbial protein are also supplements of animal protein, and these proteins have lower digestibility, and the specific digestibility values are also listed in part. All things considered, "Amino Acid Supplements function" is a more appropriate title for this section.

Question 3: 3.1. Protein Charge - in my opinion, the next chapter is not fully titled - the title indicates the types of proteins, their origin, and the content concerns rather interactions and changes depending on physicochemical factors - temperature, pH

Response 3: Dear reviewer, thank you for your kind suggestion. Based on your suggestion and reviewing the content, we have changed the title of section 3.1 to "pH of the printing inks".

Question 4: 3.4. temperature - giving such a title indicates some border values of changes that occur in various proteins, which is quite general. The chapter is too generally written in analytical and temperature processes, there is no concept of "lower", "higher" temperature, if it is higher / lower but than a specific one, e.g. 40 degrees C.

Response 4: Dear reviewer, thank you for this good advice. Based on your suggestion and reviewing the content, we found that temperature, as a title, was not appropriate in this section. Because the entire paragraph describes the physical or chemical properties of proteins after being heated, such as microwave heating or other heat treatments, and the cited literature rarely mentions specific heating temperature values, but mentions some heating parameters, such as microwave power, etc., so we think it is more appropriate to change the title of this section to "Heating treatment".

Question 5: Suggests for the parentage 2.2. and consider all 3 to give new titles more relevant

Response 5: Thank you for this good advice. We have listened to your suggestion in question 2 and have revised the title and content of this section. There is no need for a multi-faceted discussion of this section. The writing of this part mainly focuses on the complementation effect of the amino acid diversity of proteins on the body, the composition and structure of proteins from different sources are diverse. Therefore, it is more appropriate to concentrate this part on the supplementation of amino acids in proteins to the human body. Thank you for helping us to improve the quality of our review article.
